# Multimorbidity, mortality, and HbA1c in type 2 diabetes: A cohort study with UK and Taiwanese cohorts

**Jason I. Chiang** [1]*, **Peter Hanlon** [2], **Tsai-Chung Li** [3], **Bhautesh Dinesh Jani** [2], **Jo-Anne Manski-Nankervis** [1], **John Furler** [1], **Cheng-Chieh Lin** [4], **Shing-Yu Yang** [3], **Barbara I. Nicholl** [2], **Sharmala Thuraisingam** [1], **Frances S. Mair** [2]

1 Department of General Practice, University of Melbourne, Melbourne, Australia, 2 General Practice and Primary Care, Institute of Health and Wellbeing, University of Glasgow, Glasgow, United Kingdom,
3 Department of Public Health, College of Public Health, China Medical University, Taichung, Taiwan,
4 Department of Family Medicine, China Medical University Hospital, Taichung, Taiwan

* jason.chiang@unimelb.edu.au

**Data Availability Statement:** All relevant data are within the manuscript and its Supporting Information files.

## Abstract

### Background

There is emerging interest in multimorbidity in type 2 diabetes (T2D), which can be either concordant (T2D related) or discordant (unrelated), as a way of understanding the burden of disease in T2D. Current diabetes guidelines acknowledge the complex nature of multimorbidity, the management of which should be based on the patient's individual clinical needs and comorbidities. However, although associations between multimorbidity, glycated haemoglobin (HbA1c), and mortality in people with T2D have been studied to some extent, significant gaps remain, particularly regarding different patterns of multimorbidity, including concordant and discordant conditions. This study explores associations between multimorbidity (total condition counts/concordant/discordant/different combinations of conditions), baseline HbA1c, and all-cause mortality in T2D.

### Methods and findings

We studied two longitudinal cohorts of people with T2D using the UK Biobank ($n = 20,569$) and the Taiwan National Diabetes Care Management Program (NDCMP) ($n = 59,657$). The number of conditions in addition to T2D was used to quantify total multimorbidity, concordant, and discordant counts, and the effects of different combinations of conditions were also studied. Outcomes of interest were baseline HbA1c and all-cause mortality. For the UK Biobank and Taiwan NDCMP, mean (SD) ages were 60.2 (6.8) years and 60.8 (11.3) years; 7,579 (36.8%) and 31,339 (52.5%) were female; body mass index (BMI) medians (IQR) were 30.8 (27.7, 34.8) kg/m² and 25.6 (23.5, 28.7) kg/m²; and 2,197 (10.8%) and 9,423 (15.8) were current smokers, respectively. Increasing total and discordant multimorbidity counts were associated with lower HbA1c and increased mortality in both datasets. In Taiwan NDCMP, for those with four or more additional conditions compared with T2D only, the mean difference (95% CI) in HbA1c was −0.82% (−0.88, −0.76) $p < 0.001$. In UK Biobank, hazard ratios (HRs) (95% CI) for all-cause mortality in people with T2D and one, two, three,

**Funding:** The authors received no specific funding for this work.

**Competing interests:** The authors have declared that no competing interests exist.

**Abbreviations:** AF, atrial fibrillation; BMI, body mass index; CHD, coronary heart disease; CKD, chronic kidney disease; COPD, chronic obstructive pulmonary disease; HbA1c, glycated haemoglobin; HES, Hospital Episode Statistics; HF, heart failure; HR, hazard ratio; ICD, International Classification of Disease; NDCMP, National Diabetes Care Management Program; NTD, New Taiwan dollar; OAD, oral antidiabetes drug; PVD, peripheral vascular disease; T2D, type 2 diabetes; TIA, transient ischaemic attack..

and four or more additional conditions compared with those without comorbidity were 1.20 (0.91–1.56) $p < 0.001$, 1.75 (1.35–2.27) $p < 0.001$, 2.17 (1.67–2.81) $p < 0.001$, and 3.14 (2.43–4.03) $p < 0.001$, respectively. Both concordant/discordant conditions were significantly associated with mortality; however, HRs were largest for concordant conditions. Those with four or more concordant conditions had >5 times the mortality (5.83 [4.28–7.93] $p < 0.001$). HRs for NDCMP were similar to those from UK Biobank for all multimorbidity counts. For those with two conditions in addition to T2D, cardiovascular diseases featured in 18 of the top 20 combinations most highly associated with mortality in UK Biobank and 12 of the top combinations in the Taiwan NDCMP. In UK Biobank, a combination of coronary heart disease and heart failure in addition to T2D had the largest effect size on mortality, with a HR (95% CI) of 4.37 (3.59–5.32) $p < 0.001$, whereas in the Taiwan NDCMP, a combination of painful conditions and alcohol problems had the largest effect size on mortality, with an HR (95% CI) of 4.02 (3.08–5.23) $p < 0.001$. One limitation to note is that we were unable to model for changes in multimorbidity during our study period.

## Conclusions

Multimorbidity patterns associated with the highest mortality differed between UK Biobank (a population predominantly comprising people of European descent) and the Taiwan NDCMP, a predominantly ethnic Chinese population. Future research should explore the mechanisms underpinning the observed relationship between increasing multimorbidity count and reduced HbA1c alongside increased mortality in people with T2D and further examine the implications of different patterns of multimorbidity across different ethnic groups. Better understanding of these issues, especially effects of condition type, will enable more effective personalisation of care.

## Author summary

### Why was this study done?

- People with type 2 diabetes (T2D) commonly have other coexisting chronic medical conditions ('multimorbidity'). These conditions can be either concordant (T2D related) or discordant (T2D unrelated).

- Multimorbidity is associated with higher mortality and hypoglycaemia; however, the effect of multimorbidity on glycaemia (measured by glycated haemoglobin [HbA1c]) is mixed.

- Significant knowledge gaps remain, particularly regarding the prevalence and impacts of different patterns of multimorbidity, including concordant and discordant conditions, and their associations with HbA1c and mortality.

### What did the researchers do and find?

- We assessed the associations between different counts of multimorbidity, including concordant and discordant conditions, and HbA1c and the effects of different combinations of conditions on all-cause mortality in people with T2D.

- In two large community cohorts of people with T2D (UK Biobank and Taiwan National Diabetes Care Management Program [NDCMP]), we found that increasing multimorbidity is significantly associated with increased mortality and with lower HbA1c.

- The combinations of conditions with the greatest association with mortality differed between UK Biobank, a population predominantly comprising people of European descent, and the Taiwan NDCMP, a predominantly ethnic Chinese population.

### What do these findings mean?

- To our knowledge, this is the first study to assess and compare the relationship between total, concordant, and discordant multimorbidity counts; HbA1c; and all-cause mortality in people with T2D or to look at the effects of such a range of combinations of comorbid conditions.

- Our findings suggest the need for further research to explore the effects of different combinations of conditions on outcomes in those with T2D across different ethnic groups.

- Our findings suggest that poor glycaemic control is unlikely to explain the increased mortality seen in those with increasing multimorbidity count.

## Introduction

Multimorbidity, the presence of two or more chronic conditions [1], is the norm in people with type 2 diabetes (T2D). Approximately 85% of people with T2D have at least one other chronic condition [2], making multimorbidity in this population an important clinical and public health priority. Multimorbidity brings many challenges, including difficulties in managing the competing demands of multiple conditions. Self-management of any chronic condition can be burdensome, and those with multimorbidity are likely to experience greater levels of treatment burden because of the complex self-management requirements imposed by different conditions [3]. This can result in reduced adherence to complicated therapeutic regimens and poorer outcomes [4]. For people with T2D, this could lead to suboptimal glycaemic management, which has been shown to result in the development of complications and increased mortality [5,6].

Currently, there is no universally accepted measure of multimorbidity. Despite this, studies using a range of different methodologies in varying study settings consistently show that multimorbidity in people with T2D is associated with higher risk of death [7]. However, only one study attempted to assess the influence of type of condition included in their multimorbidity count by differentiating between physical and mental health condition counts [8]. This resonates with the wider multimorbidity literature, which suggests that type as well as number of conditions is important [1]. Piette and Kerr have suggested that multiple conditions in those with T2D should be qualitatively assessed as concordant or discordant [5]. Concordant conditions are closely related to T2D and represent parts of the same overall pathophysiologic risk profile and are more likely to be the focus of the same disease and management plan (e.g., hypertension), whereas discordant conditions are not directly related in either their pathogenesis or management (e.g., depression, osteoarthritis, and cancer) [5].

Although the associations between multimorbidity, glycated haemoglobin (HbA1c) [7,9], and all-cause mortality [7,8,10] in people with T2D have been studied to some extent,

significant gaps remain in the existing literature, particularly regarding different patterns of multimorbidity, including concordant and discordant conditions, and their associations with HbA1c and mortality. This study addresses this evidence gap and aims to assess the associations between different counts of multimorbidity, including concordant and discordant conditions, on HbA1c and all-cause mortality in people with T2D. We also aim to understand whether associations between multimorbidity and our outcomes are universally consistent across separate cohorts from two countries with different healthcare systems and differing ethnicities, using data from the UK Biobank (a large community cohort of more than half a million people across the United Kingdom) [11] and the Taiwan National Diabetes Care Management Program (NDCMP) (a large cohort of people with T2D across Taiwan) [12].

## Methods

### Study design and participants

Two large community cohorts were used in this study. The UK Biobank (described elsewhere) [11] includes 502,640 participants recruited between 2006 and 2010, with linkage to routine healthcare data until 2018. We identified 20,569 people with T2D using a published algorithm developed by Eastwood and colleagues [13], which has been validated externally using primary and secondary care hospital data linked to the UK Biobank [13]. All 20,569 people with T2D in the UK Biobank were included in the statistical analysis.

The Taiwan NDCMP (described elsewhere) [12] includes 63,084 ethnic Chinese participants with any type of diabetes recruited between 2001 and 2004 and followed until 2011. We excluded those with type 1 diabetes and gestational diabetes. In the final analysis, 59,657 people with T2D from the Taiwan NDCMP were included.

Ethics approvals were granted by the NHS National Research Ethics Service (generic ethics approval for UK Biobank studies, approval letter dated 17 June 2011, Ref 11/NW/0382), the China Medical University Hospital Ethical Review Board (CMUH106-REC1-148), and the University of Melbourne Human Research Ethics Committee (Ethic ID: 1851038.1). Our detailed study protocol is shown in S1 Text, and we have adhered to the STROBE statement (see S2 Text).

### Procedures

We classified multimorbidity on the basis of a count of 42 chronic conditions in addition to T2D based on previously published literature (see S1 Table) [1]. In the UK Biobank, conditions were identified using self-reported conditions from the nurse-led interview as well as using linkage to Hospital Episode Statistics (HES). Participants were considered to have a condition if it was either self-reported or if they had relevant International Classification of Disease (ICD)-10-CM codes from a hospital episode occurring prior to the assessment centre date. In the Taiwan NDCMP, conditions were identified using ICD-9-CM codes to search hospital data from inpatient care and outpatient visits including primary care. We qualitatively assessed each of the comorbid conditions and categorised these as either concordant or discordant based on Piette and Kerr's [5] definitions mentioned previously. We presented multimorbidity in three ways: total, concordant, and discordant condition counts. Each of the counts were categorised into zero, one, two, three, or four or more conditions (in addition to T2D).

In both datasets, age, body mass index (BMI), duration of diabetes, and baseline HbA1c were used as continuous variables. Sex and use of corticosteroids were used as categorical variables. Use of glucose-lowering drugs was classified into no medication, one noninsulin antidiabetic drug (oral antidiabetes drug [OAD]), two OADs, three OADs, more than three OADs, insulin only, or insulin and OAD.

In the UK Biobank, socioeconomic status was classified into quintiles based on Townsend score (an area-based measure of deprivation in the UK) on the whole UK Biobank [14]: category 1 was the least deprived, and category 5 was the most deprived category. Smoking status was classified into two categories: yes (current) or no. Alcohol intake was based on self-reported frequency of alcohol intake: never or special occasions only, one to three times per month, one to four times per week, or daily. Physical activity was self-reported based on responses from the UK Biobank physical activity questionnaire. We categorised the responses into none (no physical activity in the last 4 weeks), low (light activity [e.g., pruning, watering the lawn] only in the last 4 weeks), medium (heavy activity [e.g., weeding, lawn mowing, carpentry, and digging] and/or walking for pleasure and/or other exercises in the last 4 weeks), or high (strenuous sports in the last 4 weeks). Data on physical activity were only available in the UK Biobank dataset.

In the Taiwan NDCMP, socioeconomic status was measured by amount of health insurance premium, insured unit, and residential area. Smoking status and alcohol consumption were classified into two categories: yes (current) or no. Number of outpatient visits was used as a continuous variable and was only available in the Taiwan NDCMP dataset.

## Outcome

There were two outcomes of interest: baseline HbA1c and all-cause mortality. We explored the cross-sectional association between multimorbidity counts and the most recent HbA1c measure collected at time of recruitment. In contrast, the association between multimorbidity counts and all-cause mortality explored was longitudinal. In the UK Biobank, all-cause mortality data were from the national mortality records linked by the UK Biobank up to 2018. The median (IQR) follow-up duration was 8.8 years (97–104 months). In the Taiwan NDCMP, all participants were followed from time of entry into the study to 31 December 2011 or until death or withdrawal from the NDCMP. The median (IQR) follow-up duration was 8.8 years (98–110 months).

## Statistical analysis

Statistical analyses for the UK Biobank and the Taiwan NDCMP cohorts were conducted separately. Descriptive statistics summarised the overall characteristics of the participants and the prevalence of individual health conditions.

Multivariable linear regression models were used to compare baseline HbA1c between different categorical combinations of multimorbidity counts (total condition count, concordant condition count, discordant condition count). Adjustments were made for age, gender, BMI, smoking status, alcohol consumption, socioeconomic status, duration of diabetes, use of OADs, and use of corticosteroids.

Cumulative survival plots were used to compare cumulative survival between participants with T2D with different multimorbidity counts. This was done for total condition count, concordant condition count, and discordant condition count.

Multivariable Cox proportional hazards models were used to compare all-cause mortality between different categorical combinations of multimorbidity counts (total condition count, concordant condition count, discordant condition count). Adjustments made were the same as those described above plus baseline HbA1c.

Multivariable Cox proportional hazards models were fitted to each of the individual chronic conditions that had a prevalence of >1% in our study population to examine their association with all-cause mortality.

Multivariable Cox proportional hazards models were fitted to all possible combinations of two conditions in addition to T2D to examine their association with all-cause mortality. We present the top 20 combinations in terms of hazard ratios (HRs).

For the UK Biobank, all analyses were performed using R software (version 3.4.1). Syntax for the generation of derived variables and for the analysis used for this study were submitted to UK Biobank for record. For the Taiwan NDCMP, all analyses were performed with the SAS statistical package for Windows (version 9.3, SAS; Cary, NC, United States).

## Sensitivity analysis

For the multivariable linear regression models on HbA1c and multivariable Cox proportional hazards models on mortality, we further adjusted for variables that were only available in each of the datasets. In the UK Biobank analyses, we additionally adjusted for physical activity, and for the Taiwan NDCMP, we additionally adjusted for number of outpatient visits.

For the multivariable linear regression models on HbA1c and multivariable Cox proportional hazards models on mortality in the UK Biobank, we reran the models using only HES data to identify multimorbidity.

## Results

In total, 20,569 and 59,657 people with T2D in the UK Biobank and the Taiwan NDCMP, respectively, were included in the study for analysis. In the UK Biobank, more than 90% of participants were multimorbid (having at least one chronic condition in addition to T2D), whereas approximately 80% of those in the Taiwan NDCMP were multimorbid. Table 1 describes the overall characteristics of participants included in our study.

Table 2 shows the prevalence of individual conditions included in our multimorbidity total, concordant, and discordant counts. In the UK Biobank, 15,654 (76.1%) participants had at least one concordant condition, and 13,753 (66.9%) had at least one discordant condition in addition to T2D. In the Taiwan NDCMP, a slightly lower proportion of participants had at least one concordant condition (57.2%), and a similar proportion had at least one discordant condition compared (58.0%) with the UK Biobank. The most prevalent condition was hypertension, with a prevalence of 69.0% and 48.2%, respectively.

Table 3 shows the mean difference in HbA1c between participants with different multimorbidity counts. Participants with T2D only were the reference group. In both the UK Biobank and the Taiwan NDCMP, increasing total multimorbidity and discordant counts were associated with lower HbA1c. Notably, the mean difference in HbA1c was greater in the Taiwan NDCMP compared with the UK Biobank. For concordant conditions, associations between increasing concordant counts and lower HbA1c were only observed in the Taiwan NDCMP, whereas there was no association in the UK Biobank. In the sensitivity analysis, when physical activity was additionally adjusted for in the UK Biobank, the results were similar. However, when the number of outpatient visits was additionally adjusted for in the Taiwan NDCMP, the associations between all multimorbidity counts and HbA1c attenuated slightly (S2 Table). When we used only the HES data to identify multimorbidity in the UK Biobank, the results for associations between multimorbidity and HbA1c were similar to our main analysis (S4 Table).

Fig 1 compares the unadjusted survival among the study participants on the basis of total multimorbidity, concordant, and discordant condition counts, respectively. For a given count of concordant conditions, mortality was higher (or survival lower) than for an equivalent count of discordant conditions or any conditions. Notably, in all three counts of multimorbidity (total, concordant, and discordant) the survival rate in the Taiwan NDCMP compared with the UK Biobank was much lower, with a steeper increase in proportion of death.

**Table 1. Characteristics of participants with T2D.**

| | |
|---|---|
| **UK Biobank (N = 20,569)** | |
| **Age (years), mean (SD)** | 60.2 (6.8) |
| **Female, *n* (%)** | 7,579 (36.8) |
| **BMI (kg/m$^2$), median (IQR)** | 30.8 (27.7, 34.8) |
| Missing | 179 |
| **Smoking status, yes (current), *n* (%)** | 2,197 (10.8) |
| Missing | 223 |
| **Alcohol frequency, *n* (%)** | |
| Never | 7,154 (34.9) |
| 1–3 times per month | 2,521 (12.3) |
| 1–4 times per week | 7,851 (38.3) |
| Daily | 2,946 (14.4) |
| Missing | 97 |
| **Physical activity, *n* (%)** | |
| None | 2,830 (14.1) |
| Low | 1,209 (6.0) |
| Medium | 15,322 (76.4) |
| High | 684 (3.4) |
| Missing | 524 |
| **Baseline HbA1c (%), mean (SD)** | 6.8 (1.2) |
| Missing | 1,597 |
| **Duration of diabetes (years), median (IQR)** | 4 (2, 8) |
| **Type of glucose-lowering drug use, *n* (%)** | |
| No medication | 6,778 (33.0) |
| 1 OAD | 8,036 (39.1) |
| 2 OADs | 4,412 (21.4) |
| 3 OADs | 715 (3.5) |
| >3 OADs | 9 (0.0) |
| Insulin + OAD | 619 (3.0) |
| **Use of corticosteroids, *n* (%)** | 203 (0.0) |
| **Townsend score, *n* (%)** | |
| Category 1—least deprived | 2,993 (14.6) |
| Category 2 | 3,315 (16.1) |
| Category 3 | 3,687 (18.0) |
| Category 4 | 4,334 (21.1) |
| Category 5—most deprived | 6,209 (30.2) |
| Missing | 31 |
| **Number of chronic conditions, *n* (%)** | |
| None | 1,918 (9.3) |
| 1 | 5,114 (24.9) |
| 2 | 5,109 (24.8) |
| 3 | 3,643 (17.7) |
| ≥4 | 4,785 (23.3) |
| **Taiwan NDCMP (N = 59,657)** | |
| **Age (years), mean (SD)** | 60.8 (11.3) |
| **Female, *n* (%)** | 31,339 (52.5) |
| **BMI (kg/m$^2$), median (IQR)** | 25.6 (23.5, 28.7) |
| Missing | 1,376 |

(*Continued*)

**Table 1.** (Continued)

| UK Biobank (*N* = 20,569) | |
| --- | --- |
| **Smoking status, yes (current), *n* (%)** | 9,423 (15.8) |
| Missing | 45 |
| **Alcohol consumption, yes, *n* (%)** | 5,140 (8.6) |
| Missing | 55 |
| **Baseline HbA1c (%), mean (SD)** | 8.2 (2.0) |
| Missing | 276 |
| **Duration of diabetes (years), median (IQR)** | 5 (1, 9) |
| **Type of glucose-lowering drug use, *n* (%)** | |
| No medication | 1,880 (3.2) |
| 1 OAD | 10,714 (18.0) |
| 2 OADs | 24,401 (40.9) |
| 3 OADs | 10,361 (17.4) |
| >3 OADs | 3,000 (5.0) |
| Insulin use | 1,719 (2.9) |
| Insulin + OAD | 7,582 (12.7) |
| **Use of corticosteroids, *n* (%)** | 2,964 (5.0) |
| **Urbanisation level, *n* (%)** | |
| 1—most deprived | 12,495 (21.0) |
| 2 | 18,962 (31.9) |
| 3 | 10,504 (17.7) |
| 4 | 11,735 (19.7) |
| >4—least deprived | 5,808 (9.8) |
| Missing | 153 |
| **Amount of insured premium (NTD$ per month), median (IQR)** | 21,900 (1,317, 31,800) |
| **Insured unit, *n* (%)** | |
| Government, school, or private enterprise employees | 20,491 (34.5) |
| Member of occupational, farmers, fishermen groups | 24,625 (41.5) |
| Low-income households and veterans | 14,206 (24.0) |
| Missing | 335 |
| **Number of outpatient visits, mean (SD)** | 23.75 (15.4) |
| **Number of chronic conditions, *n* (%)** | |
| None | 12,950 (21.7) |
| 1 | 15,485 (26.0) |
| 2 | 15,139 (25.4) |
| 3 | 8,330 (14.0) |
| ≥4 | 7,753 (13.0) |

Abbreviations: BMI, body mass index; HbA1c, glycated haemoglobin; NDCMP, National Diabetes Care Management Program; NTD, New Taiwan dollar; OAD, oral antidiabetes drug; T2D, type 2 diabetes

Table 4 shows the unadjusted and adjusted HRs comparing categories of total, concordant, and discordant multimorbidity counts with T2D and all-cause mortality (with a reference group of those with T2D only). In both the UK Biobank and the Taiwan NDCMP, increasing total, concordant, and discordant multimorbidity counts were all significantly associated with increased all-cause mortality. In the UK Biobank, the HRs (95% CI) for having one, two, three, and four or more total multimorbidity conditions compared with those with T2D only were 1.20 (0.91–1.56) $p < 0.001$, 1.75 (1.35–2.27) $p < 0.001$, 2.17 (1.67–2.81) $p < 0.001$, and 3.14

**Table 2. Prevalence of individual multimorbid conditions in participants with T2D.**

| Presence of chronic conditions concordant with T2D, *n* (%) | UK Biobank (*N* = 20,569) | Taiwan NDCMP (*N* = 59,657) |
|---|---|---|
| At least one chronic condition concordant with diabetes | 15,654 (76.1) | 34,111 (57.2) |
| Hypertension | 14,187 (69.0) | 28,771 (48.2) |
| Coronary heart disease | 3,773 (18.3) | 8,639 (14.5) |
| Peripheral vascular disease | 488 (2.4) | 1,711 (2.9) |
| Chronic kidney disease | 323 (1.6) | 1,919 (3.2) |
| Stroke/TIA | 1,024 (5.0) | 4,350 (7.3) |
| Diabetic retinopathy | 2,174 (10.6) | 1,494 (2.5) |
| Diabetic neuropathy | 74 (0.4) | 642 (1.1) |
| Atrial fibrillation | 641 (3.1) | 472 (0.8) |
| Heart failure | 426 (2.1) | 1,394 (2.3) |
| **Presence of chronic conditions discordant with T2D, *n* (%)** | **UK Biobank (*N* = 20,569)** | **Taiwan NDCMP (*N* = 59,657)** |
| At least one chronic condition discordant with diabetes | 13,753 (66.9) | 34,592 (58.0) |
| Depression | 1,643 (8.0) | 628 (1.1) |
| Painful conditions (excluding diabetic neuropathy) | 6,250 (30.4) | 13,754 (23.1) |
| Asthma | 2,959 (14.4) | 1,404 (2.4) |
| Dyspepsia | 3,815 (18.5) | 12,297 (20.6) |
| Thyroid disorders | 1,688 (8.2) | 1,618 (2.7) |
| Rheumatoid arthritis and other connective tissue disorders | 618 (3.0) | 309 (0.5) |
| COPD | 841 (4.1) | 4,000 (6.7) |
| Anxiety | 509 (2.5) | 3,398 (5.7) |
| Irritable bowel syndrome | 500 (2.4) | 636 (1.1) |
| Cancer | 2,110 (10.3) | 1,107 (1.9) |
| Alcohol problems | 427 (2.1) | 370 (0.6) |
| Other psychoactive substance misuse | 13 (0.1) | 23 (0.0) |
| Constipation | 288 (1.4) | 2,670 (4.5) |
| Diverticular disease | 1,056 (5.1) | 86 (0.1) |
| Prostate disorders | 890 (4.3) | 2,513 (4.2) |
| Glaucoma | 458 (2.2) | 305 (0.5) |
| Epilepsy | 211 (1.0) | 135 (0.2) |
| Dementia | 10 (0.0) | 330 (0.6) |
| Schizophrenia/bipolar disorder | 187 (0.9) | 380 (0.6) |
| Psoriasis/eczema | 792 (3.9) | 1,704 (2.9) |
| Inflammatory bowel disease | 924 (4.5) | 24 (0.0) |
| Migraine | 306 (1.5) | 162 (0.3) |
| Chronic sinusitis | 176 (0.9) | 152 (0.3) |
| Anorexia/bulimia | 2 (0.0) | 148 (0.3) |
| Bronchiectasis | 56 (0.3) | 162 (0.3) |
| Parkinson disease | 42 (0.2) | 309 (0.5) |
| Multiple sclerosis | 71 (0.3) | 4 (0.0) |
| Viral hepatitis | 56 (0.3) | 1,263 (2.1) |
| Chronic liver disease | 326 (1.6) | 7,047 (11.8) |
| Osteoporosis | 340 (1.7) | 1,322 (2.2) |
| Chronic fatigue syndrome | 71 (0.3) | 0 (0.0) |
| Endometriosis | 162 (0.8) | 100 (0.2) |

(*Continued*)

**Table 2.** (Continued)

| Meniere disease | 52 (0.3) | 493 (0.8) |
| Pernicious anaemia | 134 (0.7) | 31 (0.1) |
| Polycystic ovary | 31 (0.2) | 17 (0.0) |

Abbreviations: COPD, chronic obstructive pulmonary disease; NDCMP, National Diabetes Care Management Program; T2D, type 2 diabetes; TIA, transient ischaemic attack

**Table 3. Multivariable linear regression model: Relationship between HbA1c and multimorbidity in participants with type 2 diabetes.**

| | UK Biobank | | | | | Taiwan NDCMP | | | | |
|---|---|---|---|---|---|---|---|---|---|---|
| **Predictor variables** | | **Unadjusted** | | **Adjusted***  | | | **Unadjusted** | | **Adjusted***  | |
| **Categories of diabetes and multimorbidity** | **Deaths/ N** | **Mean difference in HbA1c (95% CI)** | **P value** | **Mean difference in HbA1c (95% CI)** | **P value** | **Deaths/N** | **Mean difference in HbA1c (95% CI)** | **P value** | **Mean difference in HbA1c (95% CI)** | **P value** |
| Diabetes only (reference) | 79/1,918 | Ref | | Ref | | 1,493/ 12,950 | Ref | | Ref | |
| Diabetes plus 1 chronic condition | 280/ 5,114 | −0.07 (−0.14, −0.01) | 0.024 | −0.07 (−0.13, −0.01) | 0.031 | 2,430/ 15,480 | −0.49 (−0.54, −0.45) | <0.001 | −0.62 (−0.67, −0.58) | <0.001 |
| Diabetes plus 2 chronic conditions | 421/ 5,109 | −0.13 (−0.19, −0.06) | <0.001 | −0.12 (−0.18, −0.06) | <0.001 | 3,318/ 15,139 | −0.57 (−0.61, −0.52) | <0.001 | −0.72 (−0.76, −0.67) | <0.001 |
| Diabetes plus 3 chronic conditions | 395/ 3,643 | −0.11 (−0.17, −0.04) | 0.002 | −0.13 (−0.19, −0.06) | <0.001 | 2,505/ 8,330 | −0.55 (−0.61, −0.49) | <0.001 | −0.75 (−0.80, −0.69) | <0.001 |
| Diabetes plus ≥4 chronic conditions | 759/ 4,785 | −0.16 (−0.23, −0.10) | <0.001 | −0.20 (−0.26, −0.14) | <0.001 | 3,477/ 7,753 | −0.55 (−0.60, −0.49) | <0.001 | −0.82 (−0.88, −0.76) | <0.001 |
| **Categories of diabetes and concordant conditions** | | | | | | | | | | |
| Diabetes only (reference) | 79/1,918 | Ref | | Ref | | 1,493/ 12,950 | Ref | | Ref | |
| Diabetes plus 1 concordant condition | 768/ 10,251 | −0.14 (−0.20, −0.08) | <0.001 | −0.11 (−0.18, −0.06) | <0.001 | 5,043/ 22,407 | −0.59 (−0.63, −0.54) | <0.001 | −0.70 (−0.74, −0.65) | <0.001 |
| Diabetes plus 2 concordant conditions | 507/ 3,867 | −0.06 (−0.12, 0.01) | 0.098 | −0.10 (−0.16, −0.03) | 0.003 | 2,950/ 8,865 | −0.59 (−0.65, −0.54) | <0.001 | −0.74 (−0.80, −0.69) | <0.001 |
| Diabetes plus 3 concordant conditions | 247/ 1,131 | −0.07 (−0.15, 0.02) | 0.153 | −0.13 (−0.22, −0.04) | 0.003 | 1,140/ 2,221 | −0.58 (−0.67, −0.48) | <0.001 | −0.78 (−0.87, −0.69) | <0.001 |
| Diabetes plus ≥4 concordant conditions | 129/402 | 0.05 (−0.07, 0.19) | 0.417 | −0.03 (−0.16, 0.09) | 0.608 | 427/618 | −0.38 (−0.55, −0.22) | 0.029 | −0.66 (−0.83, −0.50) | <0.001 |
| **Categories of diabetes and discordant conditions** | | | | | | | | | | |
| Diabetes only (reference) | 79/1,918 | Ref | | Ref | | 1,493/ 12,950 | Ref | | Ref | |
| Diabetes plus 1 discordant condition | 574/ 6,387 | −0.11 (−0.18, −0.06) | <0.001 | −0.11 (−0.17, −0.04) | <0.001 | 4,187/ 19,177 | −0.52 (−0.57, −0.48) | <0.001 | −0.68 (−0.73, −0.63) | <0.001 |
| Diabetes plus 2 discordant conditions | 393/ 3,793 | −0.14 (−0.20, −0.07) | <0.001 | −0.14 (−0.20, −0.08) | <0.001 | 2,678/ 9,615 | −0.52 (−0.58, −0.47) | <0.001 | −0.73 (−0.78, −0.67) | <0.001 |
| Diabetes plus 3 discordant conditions | 246/ 1,951 | −0.22 (−0.30, −0.14) | <0.001 | −0.20 (−0.28, −0.13) | <0.001 | 1,335/ 3,770 | −0.54 (−0.62, −0.47) | <0.001 | −0.81 (−0.89, −0.74) | <0.001 |
| Diabetes plus ≥4 discordant conditions | 271/ 1,621 | −0.19 (−0.27, −0.11) | <0.001 | −0.23 (−0.31, −0.15) | <0.001 | 945/ 2,030 | −0.48 (−0.57, −0.38) | <0.001 | −0.79 (−0.89, −0.70) | <0.001 |

*Adjusting for age, gender, BMI, smoking status, alcohol consumption, socioeconomic status, duration of diabetes, use of oral antidiabetes drugs, and use of corticosteroids.

Abbreviations: BMI, body mass index; HbA1c, glycated haemoglobin; NDCMP, National Diabetes Care Management Program; Ref, reference

(2.43–4.03) $p < 0.001$, respectively. The HRs for the Taiwan NDCMP were slightly lower yet still statistically significant. For the Taiwan NDCMP, the HRs (95% CI) for having one, two, three, and four or more total multimorbidity conditions compared with those with T2D only were 1.17 (1.09–1.25) $p < 0.001$, 1.39 (1.30–1.48) $p < 0.001$, 1.79 (1.67–1.92) $p < 0.001$, and 2.50 (2.34–2.67) $p < 0.001$, respectively.

For concordant conditions, the HRs were larger compared with the HRs observed for total multimorbidity counts. In the UK Biobank, the HRs (95% CI) for having one, two, three, and four or more concordant conditions compared with those with T2D were 1.58 (1.23–2.03) $p < 0.001$, 2.41 (1.87–3.13) $p < 0.001$, 4.00 (3.03–5.27) $p < 0.001$, and 5.83 (4.28–7.93) $p < 0.001$, respectively. Again, the HRs for the Taiwan NDCMP were slightly lower yet still statistically significant. For the Taiwan NDCMP, the HRs (95% CI) for having one, two, three, and four or more concordant conditions compared with those with T2D were 1.42 (1.33–1.51) $p < 0.001$, 1.87 (1.75–2.00) $p < 0.001$, 2.80 (2.58–3.05) $p < 0.001$, and 3.79 (3.38–4.25) $p < 0.001$, respectively.

For discordant conditions, the HRs were smaller compared with the HRs observed for both total multimorbidity and concordant counts. In the UK Biobank, the HRs (95% CI) for having one, two, three, and four or more discordant conditions were 1.89 (1.46–2.43) $p < 0.001$, 2.09 (1.61–2.71) $p < 0.001$, 2.58 (1.95–3.40) $p < 0.001$, and 3.50 (2.65–4.62) $p < 0.001$, respectively. The HRs for the Taiwan NDCMP were slightly lower yet still statistically significant. For the Taiwan NDCMP, the HRs (95% CI) for having one, two, three, and four or more discordant conditions were 1.42 (1.33–1.51) $p < 0.001$, 1.64 (1.54–1.76) $p < 0.001$, 2.05 (1.89–2.22) $p < 0.001$, and 2.62 (2.40–2.86) $p < 0.001$, respectively.

For all the mortality analyses for the UK Biobank, the sensitivity analysis further adjusted for physical activity, whereas in the Taiwan NDCMP, further adjustment was made for number of outpatient visits. The sensitivity analyses made little difference to the associations between all counts of multimorbidity and mortality (S3 Table). When we used only the HES data to identify multimorbidity in the UK Biobank, the results for associations between multimorbidity and all-cause mortality were similar to our main analysis (S4 Table).

Figs 2, 3, 4 and 5 compare the adjusted HRs of the presence of individual concordant (Figs 2 and 3) and discordant conditions (Figs 4 and 5) (>1% prevalence) on mortality. All concordant conditions with the exception of diabetic retinopathy (in the UK Biobank) and diabetic neuropathy (in the Taiwan NDCMP) showed significant associations with increased mortality. Presence of heart failure (HF) had the largest HR in both datasets, with HR (95% CI) of 3.24 (2.69–3.91) $p < 0.001$ in the UK Biobank. Following HF, the presence of peripheral vascular disease (PVD), chronic kidney disease (CKD), and atrial fibrillation (AF) is associated with greater than 2-fold the risk of mortality in the UK Biobank. Similarly, both HF and CKD are also associated with greater than 2-fold the risk of mortality in the Taiwan NDCMP.

Among discordant conditions, presence of alcohol problems (alcohol dependency, alcoholic liver disease/alcoholic cirrhosis) (HR 2.58, 95% CI 2.07–3.21, $p < 0.001$), chronic liver disease (HR 2.29, 95% CI 1.75–3.01, $p < 0.001$), chronic obstructive pulmonary disease (COPD) (HR 2.00, 95% CI 1.69–2.36, $p < 0.001$), and cancer (HR 1.79, 95% CI 1.58–2.04, $p < 0.001$) had the largest HRs in the UK Biobank, all of which were associated with greater than 1.5-fold the risk of mortality, whereas presence of cancer (HR 2.25, 95% CI 2.06–2.45, $p < 0.001$) and viral hepatitis (HR 2.00, 95% CI 1.81–2.20, $p < 0.001$) had the largest HRs in the Taiwan NDCMP.

Figs 6 and 7 compare the adjusted HRs of the presence of the top 20 combinations (by effect size) of two conditions and all-cause mortality in the UK Biobank (Fig 6) and the Taiwan NDCMP (Fig 7). Cardiovascular diseases are present in 18 of the top 20 combinations on mortality in the UK Biobank, whereas in the Taiwan NDCMP, cardiovascular diseases are present

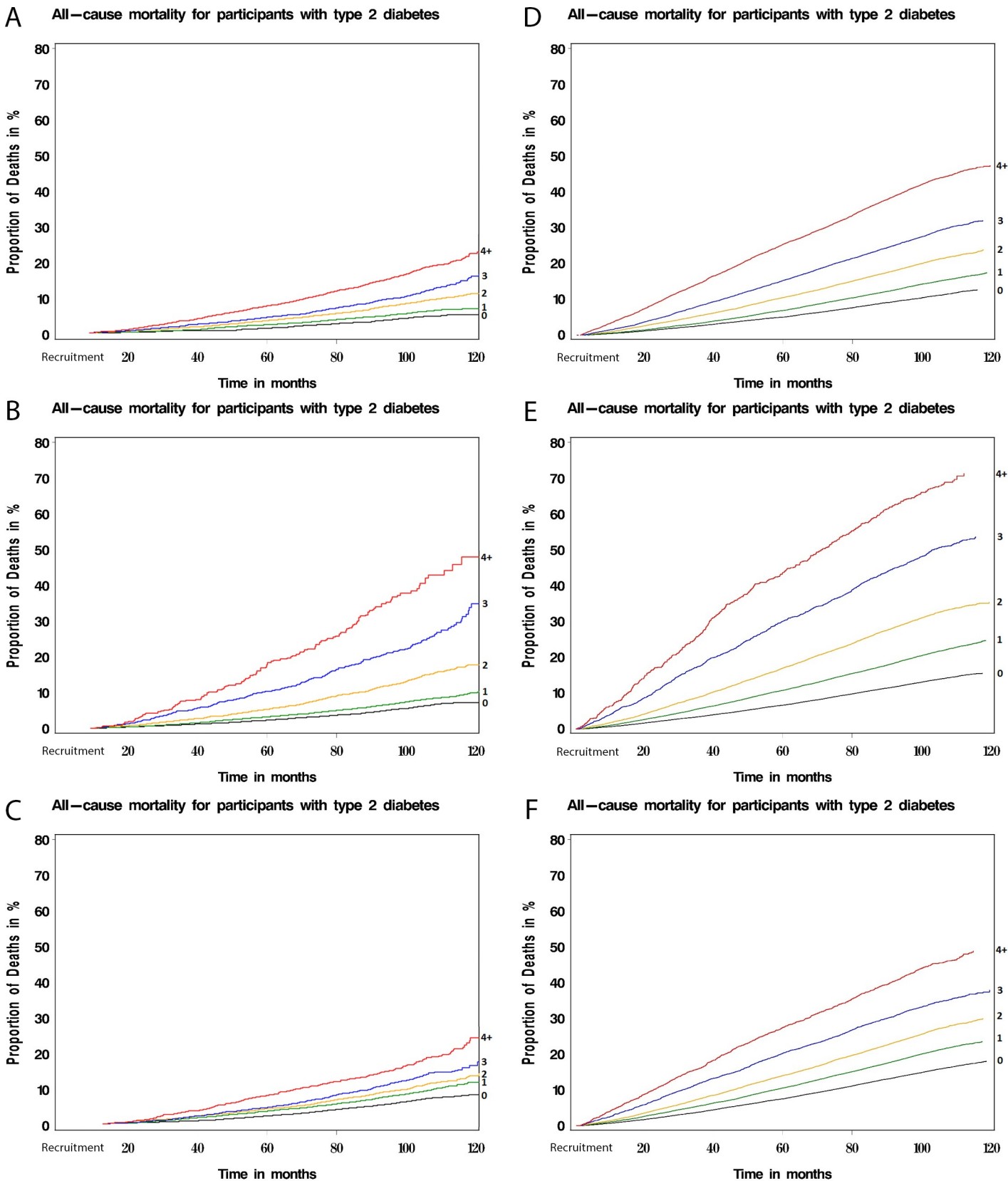

**Fig 1. Cumulative survival plot showing probability of all-cause mortality among type 2 diabetes participants with different levels of multimorbidity.** (A) Total multimorbid conditions in UK Biobank; (B) concordant conditions in UK Biobank; (C) discordant conditions in UK Biobank; (D) total multimorbid conditions in Taiwan NDCMP; (E) concordant conditions in Taiwan NDCMP; (F) discordant conditions in Taiwan NDCMP. NDCMP, National Diabetes Care Management Program.

in 12 of the top 20 combinations on mortality. In the UK Biobank, a combination of coronary heart disease (CHD) and HF had the largest effect size on mortality, with HR (95% CI) of 4.37 (3.59–5.32) $p < 0.001$. Following the combination of CHD and HF, the CHD–CKD and HF–dyspepsia combinations are each associated with greater than 4-fold the risk of mortality in the UK Biobank. The remaining top 20 combinations of conditions in the UK Biobank had greater

**Table 4. Cox's proportional hazards model: Relationship between all-cause mortality and multimorbidity in participants with type 2 diabetes.**

| Predictor variables | UK Biobank | | | | | Taiwan NDCMP | | | | |
|---|---|---|---|---|---|---|---|---|---|---|
| | | Unadjusted | | Adjusted* | | | Unadjusted | | Adjusted* | |
| Categories of diabetes and multimorbidity | Deaths/N | HRs (95% CI) | P value | HRs (95% CI) | P value | Deaths/N | HRs (95% CI) | P value | HRs (95% CI) | P value |
| Diabetes only (reference) | 79/1,918 | 1 | | 1 | | 1,493/12,950 | 1 | | 1 | |
| Diabetes plus 1 chronic condition | 280/5,114 | 1.33 (1.04–1.71) | 0.024 | 1.20 (0.91–1.56) | <0.001 | 2,430/15,480 | 1.38 (1.29–1.47) | <0.001 | 1.17 (1.09–1.25) | <0.001 |
| Diabetes plus 2 chronic conditions | 421/5,109 | 2.04 (1.61–2.60) | <0.001 | 1.75 (1.35–2.27) | <0.001 | 3,318/15,139 | 2.00 (1.88–2.13) | <0.001 | 1.39 (1.30–1.48) | <0.001 |
| Diabetes plus 3 chronic conditions | 395/3,643 | 2.73 (2.14–3.48) | <0.001 | 2.17 (1.67–2.81) | <0.001 | 2,505/8,330 | 2.90 (2.72–3.01) | <0.001 | 1.79 (1.67–1.92) | <0.001 |
| Diabetes plus ≥4 chronic conditions | 759/4,785 | 4.14 (3.28–5.22) | <0.001 | 3.14 (2.43–4.03) | <0.001 | 3,477/7,753 | 4.89 (4.60–5.20) | <0.001 | 2.50 (2.34–2.67) | <0.001 |
| **Categories of diabetes and concordant conditions** | | | | | | | | | | |
| Diabetes only (reference) | 79/1,918 | 1 | | 1 | | 1,493/12,950 | 1 | | 1 | |
| Diabetes plus 1 concordant condition | 768/10,251 | 1.84 (1.46–2.32) | <0.001 | 1.58 (1.23–2.03) | <0.001 | 5,043/22,407 | 2.06 (1.95–2.18) | <0.001 | 1.42 (1.33–1.51) | <0.001 |
| Diabetes plus 2 concordant conditions | 507/3,867 | 3.35 (2.64–4.24) | <0.001 | 2.41 (1.87–3.13) | <0.001 | 2,950/8,865 | 3.29 (3.09–3.50) | <0.001 | 1.87 (1.75–2.00) | <0.001 |
| Diabetes plus 3 concordant conditions | 247/1,131 | 5.90 (4.58–7.61) | <0.001 | 4.00 (3.03–5.27) | <0.001 | 1,140/2,221 | 5.94 (5.50–6.42) | <0.001 | 2.80 (2.58–3.05) | <0.001 |
| Diabetes plus ≥4 concordant conditions | 129/402 | 9.51 (7.18–12.58) | <0.001 | 5.83 (4.28–7.93) | <0.001 | 427/618 | 9.83 (8.83–10.95) | <0.001 | 3.79 (3.38–4.25) | <0.001 |
| **Categories of diabetes and discordant conditions** | | | | | | | | | | |
| Diabetes only (reference) | 79/1,918 | 1 | | 1 | | 1,493/12,950 | 1 | | 1 | |
| Diabetes plus 1 discordant condition | 574/6,387 | 2.23 (1.77–2.83) | <0.001 | 1.89 (1.46–2.43) | <0.001 | 4,187/19,177 | 1.99 (1.88–2.11) | <0.001 | 1.42 (1.33–1.51) | <0.001 |
| Diabetes plus 2 discordant conditions | 393/3,793 | 2.62 (2.06–3.33) | <0.001 | 2.09 (1.61–2.71) | <0.001 | 2,678/9,615 | 2.64 (2.48–2.81) | <0.001 | 1.64 (1.54–1.76) | <0.001 |
| Diabetes plus 3 discordant conditions | 246/1,951 | 3.22 (2.50–4.14) | <0.001 | 2.58 (1.95–3.40) | <0.001 | 1,335/3,770 | 3.59 (3.34–3.87) | <0.001 | 2.05 (1.89–2.22) | <0.001 |
| Diabetes plus ≥4 discordant conditions | 271/1,621 | 4.39 (3.42–5.64) | <0.001 | 3.50 (2.65–4.62) | <0.001 | 945/2,030 | 5.17 (4.76–5.61) | <0.001 | 2.62 (2.40–2.86) | <0.001 |

*Adjusting for age, gender, BMI, smoking status, alcohol consumption, socioeconomic status, baseline HbA1c, duration of diabetes, use of oral antidiabetes drugs, and use of corticosteroids.

Abbreviations: BMI, body mass index; HbA1c, glycated haemoglobin; HR, hazard ratio; NDCMP, National Diabetes Care Management Program

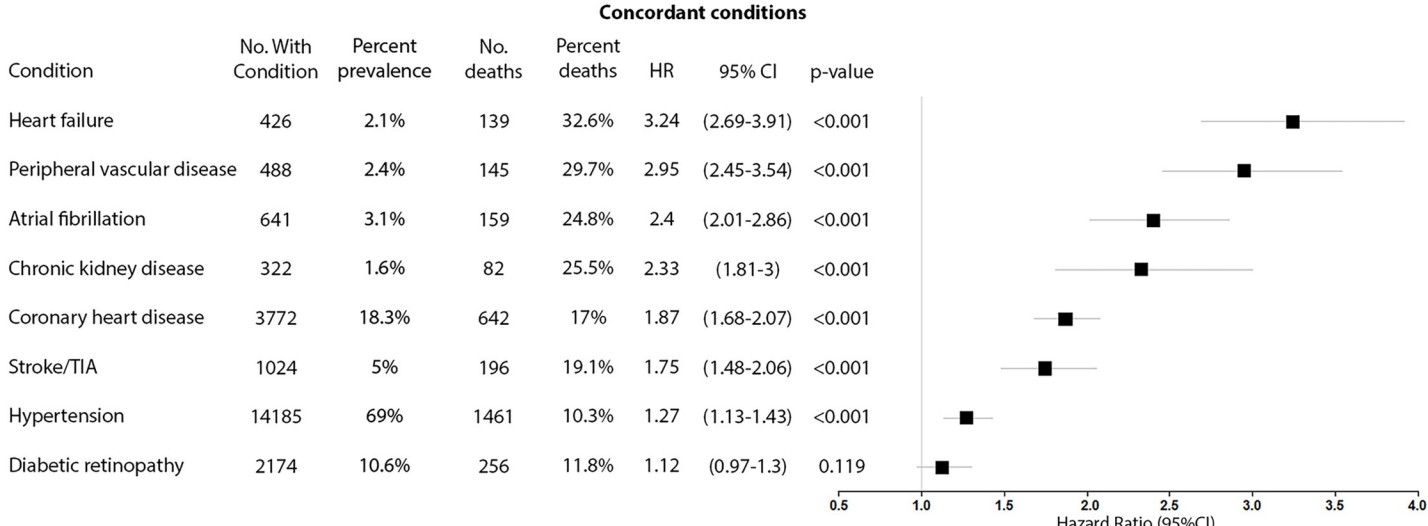

**Concordant conditions**

| Condition | No. With Condition | Percent prevalence | No. deaths | Percent deaths | HR | 95% CI | p-value |
|---|---|---|---|---|---|---|---|
| Heart failure | 426 | 2.1% | 139 | 32.6% | 3.24 | (2.69-3.91) | <0.001 |
| Peripheral vascular disease | 488 | 2.4% | 145 | 29.7% | 2.95 | (2.45-3.54) | <0.001 |
| Atrial fibrillation | 641 | 3.1% | 159 | 24.8% | 2.4 | (2.01-2.86) | <0.001 |
| Chronic kidney disease | 322 | 1.6% | 82 | 25.5% | 2.33 | (1.81-3) | <0.001 |
| Coronary heart disease | 3772 | 18.3% | 642 | 17% | 1.87 | (1.68-2.07) | <0.001 |
| Stroke/TIA | 1024 | 5% | 196 | 19.1% | 1.75 | (1.48-2.06) | <0.001 |
| Hypertension | 14185 | 69% | 1461 | 10.3% | 1.27 | (1.13-1.43) | <0.001 |
| Diabetic retinopathy | 2174 | 10.6% | 256 | 11.8% | 1.12 | (0.97-1.3) | 0.119 |

**Fig 2. Forest plot of HR for the presence of different concordant conditions (prevalence >1%) and all-cause mortality in participants with type 2 diabetes in UK Biobank.** HR, hazard ratio; No., number; TIA, transient ischaemic attack.

than 2.5-fold risk of mortality. In the Taiwan NDCMP, a combination of painful conditions and alcohol problems had the largest effect size on mortality, with HR (95% CI) of 4.02 (3.08–5.23) $p < 0.001$. Following that, combinations of dyspepsia and alcohol problems, cancer and chronic liver disease, alcohol problems and chronic liver disease, and HF and asthma are associated with greater than 3-fold the risk of mortality. The remaining top 20 combinations of conditions in the Taiwan NDCMP had greater than 2-fold risk of mortality.

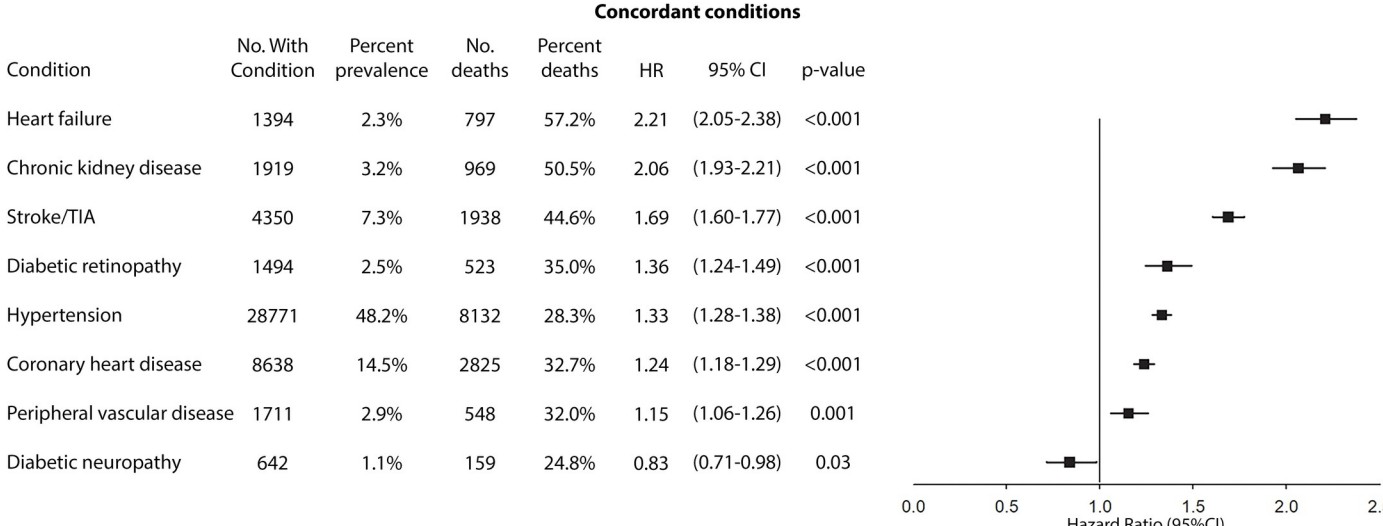

**Concordant conditions**

| Condition | No. With Condition | Percent prevalence | No. deaths | Percent deaths | HR | 95% CI | p-value |
|---|---|---|---|---|---|---|---|
| Heart failure | 1394 | 2.3% | 797 | 57.2% | 2.21 | (2.05-2.38) | <0.001 |
| Chronic kidney disease | 1919 | 3.2% | 969 | 50.5% | 2.06 | (1.93-2.21) | <0.001 |
| Stroke/TIA | 4350 | 7.3% | 1938 | 44.6% | 1.69 | (1.60-1.77) | <0.001 |
| Diabetic retinopathy | 1494 | 2.5% | 523 | 35.0% | 1.36 | (1.24-1.49) | <0.001 |
| Hypertension | 28771 | 48.2% | 8132 | 28.3% | 1.33 | (1.28-1.38) | <0.001 |
| Coronary heart disease | 8638 | 14.5% | 2825 | 32.7% | 1.24 | (1.18-1.29) | <0.001 |
| Peripheral vascular disease | 1711 | 2.9% | 548 | 32.0% | 1.15 | (1.06-1.26) | 0.001 |
| Diabetic neuropathy | 642 | 1.1% | 159 | 24.8% | 0.83 | (0.71-0.98) | 0.03 |

**Fig 3. Forest plot of HR for the presence of different concordant conditions (prevalence >1%) and all-cause mortality in participants with type 2 diabetes in Taiwan NDCMP.** HR, hazard ratio; NDCMP, National Diabetes Care Management Program; No., number; TIA, transient ischaemic attack.

**Concordant conditions**

| Condition | No. With Condition | Percent prevalence | No. deaths | Percent deaths | HR | 95% CI | p-value |
|---|---|---|---|---|---|---|---|
| Alcohol problems | 427 | 2.1% | 110 | 25.8% | 2.58 | (2.07-3.21) | <0.001 |
| Chronic Liver disease | 326 | 1.6% | 76 | 23.3% | 2.29 | (1.75-3.01) | <0.001 |
| COPD | 841 | 4.1% | 190 | 22.6% | 2 | (1.69-2.36) | <0.001 |
| Cancer | 2110 | 10.3% | 345 | 16.4% | 1.79 | (1.58-2.04) | <0.001 |
| Inflammatory bowel disease | 924 | 4.5% | 128 | 13.9% | 1.56 | (1.28-1.89) | <0.001 |
| Depression | 1643 | 8% | 198 | 12.1% | 1.41 | (1.2-1.66) | <0.001 |
| Anxiety | 509 | 2.5% | 60 | 11.8% | 1.39 | (1.05-1.84) | 0.002 |
| Psoriasis/eczema | 792 | 3.9% | 92 | 11.6% | 1.32 | (1.06-1.65) | 0.012 |
| Dyspepsia | 3815 | 18.5% | 455 | 11.9% | 1.3 | (1.16-1.46) | <0.001 |
| Constipation | 288 | 1.4% | 37 | 12.8% | 1.25 | (0.86-1.82) | 0.241 |
| Thyroid disorders | 1687 | 8.2% | 160 | 9.5% | 1.24 | (1.04-1.49) | 0.015 |
| Asthma | 2959 | 14.4% | 315 | 10.6% | 1.22 | (1.07-1.39) | 0.003 |
| Rheumatoid arthritis and other connective tissue disorders | 618 | 3% | 79 | 12.8% | 1.15 | (0.89-1.5) | 0.286 |
| Glaucoma | 458 | 2.2% | 55 | 12% | 1.14 | (0.85-1.52) | 0.389 |
| Diverticular disease | 1056 | 5.1% | 124 | 11.7% | 1.09 | (0.89-1.32) | 0.414 |
| Painful conditions | 6250 | 30.4% | 642 | 10.3% | 1.04 | (0.93-1.15) | 0.511 |
| Prostate disorders | 890 | 4.3% | 116 | 13% | 1 | (0.82-1.23) | 0.975 |
| Irritable bowel syndrome | 500 | 2.4% | 32 | 6.4% | 0.62 | (0.4-0.96) | 0.032 |
| Migraine | 306 | 1.5% | 15 | 4.9% | 0.55 | (0.31-0.97) | 0.040 |

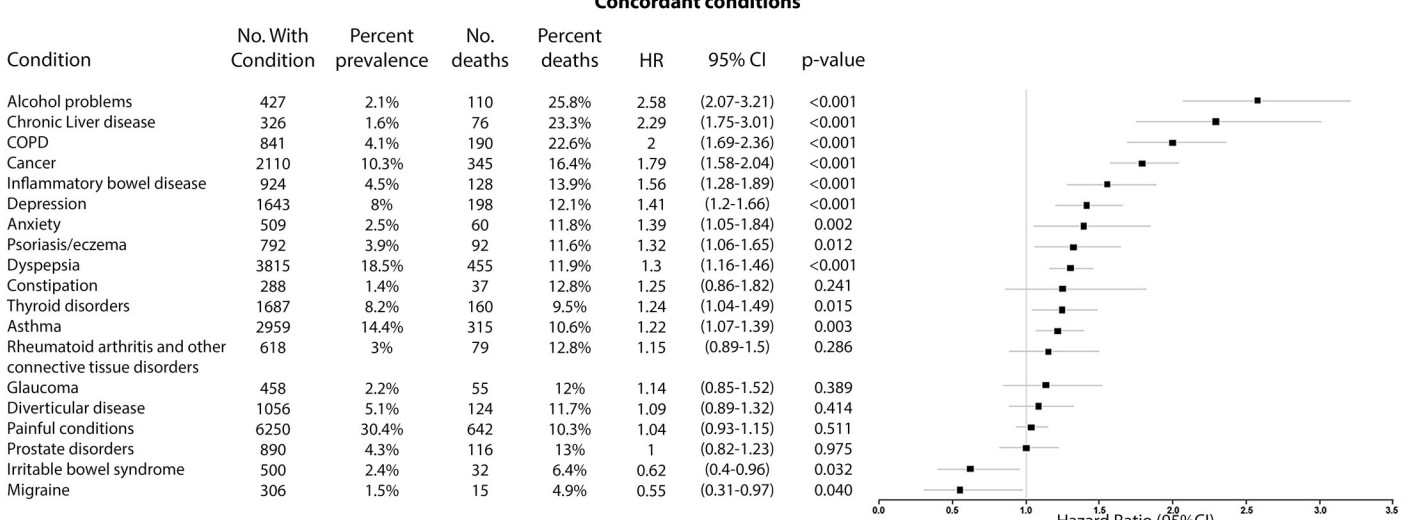

**Fig 4. Forest plot of HR for the presence of different discordant conditions (prevalence >1%) and all-cause mortality in participants with type 2 diabetes in UK Biobank.** COPD, chronic obstructive pulmonary disease; HR, hazard ratio; No., number.

## Discussion

In this study, comprising more than 80,000 middle-aged and older-aged people from two national datasets, we identified that multimorbidity was highly prevalent among people with T2D. More than 80% were found to have at least one other chronic condition in addition to T2D. We found that the associations between multimorbidity, HbA1c, and mortality are similar across two separate cohorts from two countries with different healthcare systems and differing ethnicities. Increasing total multimorbidity and discordant condition counts were associated with slightly lower HbA1c. We found significant associations between increasing multimorbidity and risk of mortality. This finding was consistent for total count of

**Concordant conditions**

| Condition | No. With Condition | Percent prevalence | No. deaths | Percent deaths | HR | 95% CI | p-value |
|---|---|---|---|---|---|---|---|
| Cancer | 1107 | 1.9% | 563 | 50.9% | 2.25 | (2.06-2.45) | <0.001 |
| Viral hepatitis | 1263 | 2.1% | 431 | 34.1% | 2.00 | (1.81-2.20) | <0.001 |
| COPD | 4000 | 6.7% | 1632 | 40.8% | 1.49 | (1.41-1.57) | <0.001 |
| Asthma | 1404 | 2.4% | 544 | 38.8% | 1.48 | (1.35-1.62) | <0.001 |
| Chronic liver disease | 7047 | 11.8% | 1785 | 25.3% | 1.42 | (1.35-1.49) | <0.001 |
| Constipation | 2670 | 4.5% | 1058 | 39.6% | 1.31 | (1.23-1.40) | <0.001 |
| Depression | 628 | 1.1% | 191 | 30.4% | 1.23 | (1.06-1.42) | 0.006 |
| Psoriasis/eczema | 1704 | 2.9% | 515 | 30.2% | 1.20 | (1.10-1.31) | <0.001 |
| Painful conditions | 13754 | 23.1% | 3798 | 27.6% | 1.15 | (1.10-1.19) | <0.001 |
| Dyspepsia | 12297 | 20.6% | 3425 | 27.9% | 1.14 | (1.09-1.18) | <0.001 |
| Prostate disorders | 2513 | 4.2% | 1001 | 39.8% | 1.06 | (0.99-1.14) | 0.10 |
| Anxiety | 3398 | 5.7% | 847 | 24.9% | 1.02 | (0.95-1.10) | 0.51 |
| Osteoporosis | 1322 | 2.2% | 349 | 26.4% | 0.99 | (0.89-1.11) | 0.89 |
| Thyroid disorders | 1618 | 2.7% | 291 | 18.0% | 0.96 | (0.85-1.08) | 0.46 |
| Irritable bowel syndrome | 636 | 1.1% | 149 | 23.4% | 0.85 | (0.72-100) | 0.06 |

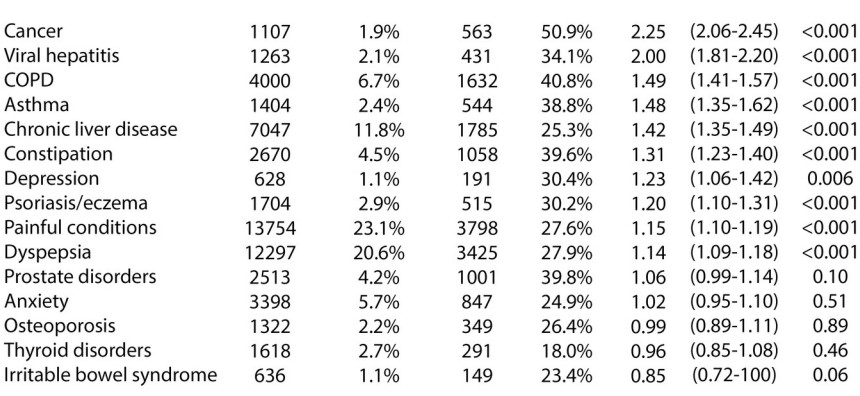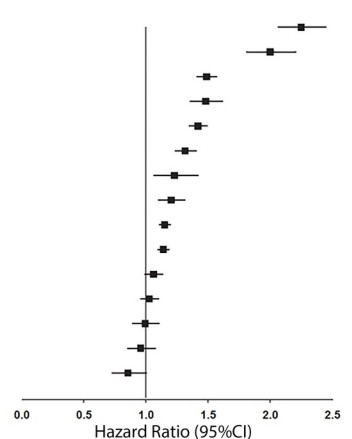

**Fig 5. Forest plot of HR for the presence of different discordant conditions (prevalence >1%) and all-cause mortality in participants with type 2 diabetes in Taiwan NDCMP.** COPD, chronic obstructive pulmonary disease; HR, hazard ratio; NDCMP, National Diabetes Care Management Program.

**Combinations of conditions: Top 20 effect sizes**

| Condition | HR | 95% CI | Total | p-value |
|---|---|---|---|---|
| Coronary heart disease and Heart Failure | 4.37 | (3.59-5.32) | 310 | <0.001 |
| Coronary heart disease and Chronic Kidney Disease | 4.29 | (3.15-5.84) | 128 | <0.001 |
| Heart Failure and Dyspepsia | 4.23 | (3.1-5.78) | 109 | <0.001 |
| Hypertension and Heart Failure | 3.88 | (3.15-4.79) | 354 | <0.001 |
| Peripheral Vascular Disease and Dyspepsia | 3.77 | (2.92-4.87) | 168 | <0.001 |
| Atrial Fibrillation and Heart Failure | 3.75 | (2.81-5) | 138 | <0.001 |
| Coronary heart disease and Peripheral Vascular Disease | 3.7 | (2.98-4.59) | 299 | <0.001 |
| Hypertension and Peripheral Vascular Disease | 3.39 | (2.77-4.16) | 413 | <0.001 |
| Coronary heart disease and Atrial Fibrillation | 3.35 | (2.73-4.11) | 340 | <0.001 |
| Coronary heart disease and COPD | 3.31 | (2.67-4.09) | 327 | <0.001 |
| Atrial Fibrillation and Dyspepsia | 3.29 | (2.48-4.37) | 159 | <0.001 |
| Hypertension and Chronic Kidney Disease | 3.27 | (2.55-4.19) | 287 | <0.001 |
| Heart Failure and Painful condition | 3.25 | (2.51-4.2) | 196 | <0.001 |
| Atrial Fibrillation and Asthma | 3.07 | (2.17-4.33) | 117 | <0.001 |
| Coronary heart disease and Stroke or TIA | 2.94 | (2.37-3.64) | 364 | <0.001 |
| Peripheral Vascular Disease and Painful condition | 2.84 | (2.17-3.73) | 195 | <0.001 |
| Chronic Kidney Disease and Painful condition | 2.73 | (1.95-3.83) | 147 | <0.001 |
| Depression and COPD | 2.65 | (1.81-3.89) | 129 | <0.001 |
| Hypertension and COPD | 2.62 | (2.16-3.17) | 647 | <0.001 |
| Hypertension and Atrial Fibrillation | 2.53 | (2.07-3.11) | 528 | <0.001 |

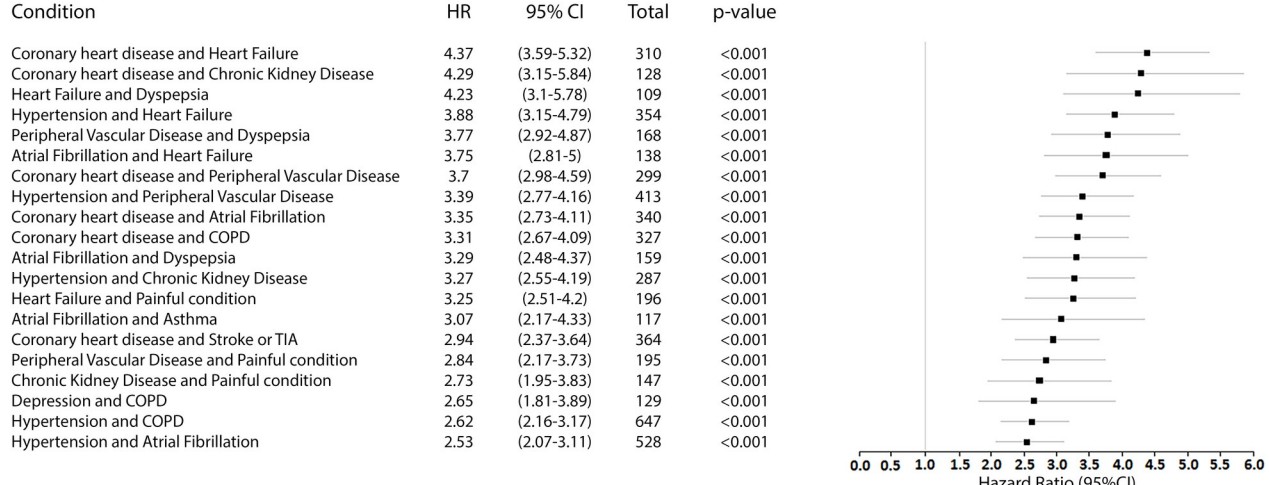

**Fig 6. Forest plot of HR for the presence of the top 20 combinations (by effect size) of two conditions and all-cause mortality in participants with type 2 diabetes in UK Biobank.** COPD, chronic obstructive pulmonary disease; HR, hazard ratio; TIA, transient ischaemic attack.

multimorbidity as well as counts of concordant and discordant conditions. This association was strongest for increasing concordant conditions. Each concordant condition, except diabetic retinopathy and neuropathy, was associated with significantly higher risk of mortality, and the presence of HF, PVD, CKD, and AF had the largest effect sizes. Presence of certain discordant conditions (alcohol problems, chronic liver disease, COPD, cancer, and viral hepatitis) had similar risk of mortality compared with concordant conditions. Our findings also show that cardiovascular diseases play a significant role in the increased mortality seen in those with multimorbidity because such conditions were in 18 of the top 20 combinations (by effect size) of two conditions in UK Biobank, whereas they were present in 12 of the top 20 combinations

**Combinations of conditions: Top 20 effect sizes**

| Condition | HR | 95% CI | Total | p-value |
|---|---|---|---|---|
| Painful conditions and Alcohol problems | 4.02 | (3.08-5.23) | 111 | <0.001 |
| Dyspepsia and Alcohol problems | 3.90 | (3.18-4.80) | 188 | <0.001 |
| Cancer and Chronic liver disease | 3.65 | (3.08-4.32) | 230 | <0.001 |
| Alcohol problems and Chronic liver disease | 3.52 | (2.97-4.18) | 314 | <0.001 |
| Heart failure and Asthma | 3.32 | (2.70-4.08) | 132 | <0.001 |
| Chronic kidney disease and Heart failure | 2.97 | (2.51-3.51) | 190 | <0.001 |
| Heart failure and COPD | 2.90 | (2.54-3.30) | 340 | <0.001 |
| COPD and Cancer | 2.73 | (2.25-3.31) | 155 | <0.001 |
| Coronary heart disease and Viral hepatitis | 2.57 | (2.06-3.20) | 170 | <0.001 |
| Dyspepsia and Viral hepatitis | 2.56 | (2.19-2.99) | 379 | <0.001 |
| Chronis kidney disease and Stroke/TIA | 2.53 | (2.22-2.89) | 352 | <0.001 |
| Stroke/TIA and Heart failure | 2.51 | (2.16-2.92) | 262 | <0.001 |
| Hypertension and Alcohol problems | 2.50 | (1.93-3.25) | 153 | <0.001 |
| COPD and Viral hepatitis | 2.50 | (1.91-3.27) | 114 | <0.001 |
| Atrial fabrillation and Heart failure | 2.46 | (2.02-2.99) | 152 | <0.001 |
| Stroke/TIA and Dementia | 2.45 | (2.01-2.98) | 155 | <0.001 |
| Chronic kidney disease and Diabetic retinopathy | 2.40 | (1.93-2.98) | 137 | <0.001 |
| Heart failure and Chronic liver disease | 2.38 | (1.91-2.97) | 153 | <0.001 |
| Coronary heart disease and Heart failure | 2.36 | (2.14-2.61) | 704 | <0.001 |
| Viral hepatitis and Chronic liver disease | 2.36 | (2.07-2.68) | 627 | <0.001 |

**Fig 7. Forest plot of HR for the presence of the top 20 combinations (by effect size) of two conditions and all-cause mortality in participants with type 2 diabetes in Taiwan NDCMP.** COPD, chronic obstructive pulmonary disease; HR, hazard ratio; NDCMP, National Diabetes Care Management Program; TIA, transient ischaemic attack.

in the Taiwan NDCMP. We have also shown that there are key differences in the top combinations (by effect size) of two conditions in addition to T2D associated with increased risk of death between the two cohorts.

Our results indicate that increasing total multimorbidity and discordant counts were associated with lower HbA1c. This adds to the existing literature and a recent systematic review, findings from which showed mixed results in terms of the associations between multimorbidity and HbA1c [7]. This is particularly interesting because it has been established that achievement of HbA1c targets is a key component of T2D management and is important in reducing all-cause mortality [15]. Although studies have also noted associations between low HbA1c and increased mortality, the causal link between the two remains unclear [16,17]. The slightly lower HbA1c observed in our study aligns with a previous study that showed that those who live with more chronic conditions receive better quality of care and higher health service use, leading to more opportunities for care of multimorbidity conditions [18]. This could also mean earlier diagnosis of the range of conditions they are living with. For example, those that have heart disease could be diagnosed earlier, meaning they are more likely to have HbA1c tests that are only mildly elevated, and subsequently receive treatment earlier. In light of this, our sensitivity analyses in the Taiwan NDCMP, in which the inclusion of the number of outpatient visits attenuated the association between multimorbidity and HbA1c, suggests that higher health service utilisation may play a role in glycaemic management in those with T2D living with multimorbidity. Furthermore, another possible explanation of the lower HbA1c seen in our results is that of survival bias. In the UK Biobank, those who live with more chronic conditions (higher degree of multimorbidity) and a higher HbA1c may not be well enough or alive to attend the baseline assessment and, hence, unable to participate in the UK Biobank; however, this would not explain the similar findings in the Taiwan dataset.

Although our results indicate associations between increasing multimorbidity and slightly lower HbA1c, a major caveat is that the degree of difference observed in HbA1c (ranging from −0.07% to −0.82%) seen in our results is not likely to be clinically significant despite being statistically significant. There have been discussions around what degree of reduction in HbA1c could be seen as clinically significant; however, it would be patient dependent [19]. Not all HbA1c improvements are equal in regard to clinical benefit; for example, a reduction of 1% in HbA1c offers different benefits if the improvement was from 12% to 11% compared with 7% to 6%. Furthermore, it is generally regarded that it is easier for patients with poorer glycaemic control to reduce their HbA1c (for example, from 10% to 9%) than for patients with relatively good glycaemic control. Although the UK Biobank cohort represents a relatively healthy population with good glycaemic control in which the mean (SD) HbA1c was 6.8 (1.2)%, it is important to note that the results were similar in the Taiwan NDCMP cohort, which is a population cohort in which the mean (SD) HbA1c was 8.2 (2.0)%.

Our study findings are consistent with previous literature in which increasing multimorbidity counts are significantly associated with higher mortality in people with T2D [7]. However, it is the first to assess the implications of concordant versus discordant multimorbidity counts and the associations between individual conditions and mortality. Only one study has assessed the type of condition in multimorbidity by differentiating between physical and mental health conditions [8]. Evidence suggests that those with more concordant conditions have improved diabetes care because of synergistic care, in which diabetes guidelines often make specific recommendations for concordant conditions but do not address discordant conditions [5,20]. As a result, to date it has been suggested that those with discordant conditions may have suboptimal care because of competition for limited resources and distraction from diabetes care, which could ultimately lead to worse outcomes and increased mortality [5,21]. Our study contributes to understanding of which patterns of multimorbidity are associated with poorer

outcomes [5,7] and demonstrates that both concordant and discordant conditions are associated with mortality but that discordant conditions generally have lower risks of death. However, we show that particular discordant conditions, such as alcohol problems, chronic liver disease, COPD, and cancer, have equal risk of mortality to concordant conditions. We also contribute to the understanding of patterns of multimorbidity that are associated with poorer outcomes in different ethnic groups through exploring associations between the top combinations of conditions and mortality in a predominantly white population (UK Biobank) and predominantly ethnic Chinese one (Taiwan NDCMP cohort). Our findings show that cardiovascular diseases are present in the majority of the top combinations that are associated with the highest risk of death in both cohorts, consistent with previous literature [22,23]. Importantly, although we have noted the significant contributions of cardiovascular diseases to increased mortality, particularly in the UK Biobank population, our findings also suggest that certain combinations of discordant conditions are also strongly associated with increased mortality. This was particularly marked in the Taiwan NDCMP cohort, in which alcohol problems, chronic liver disease, cancer, painful conditions, and dyspepsia were in the top four combinations (by effect size) of two conditions associated with increased risk of death, whereas only dyspepsia featured as part of a combination in the top five combinations in UK Biobank. This highlights the need for further research to consider the importance of ethnic differences when considering the implications of multimorbidity in people with T2D. It also underscores the importance of personalised care that takes account of individual characteristics, including ethnicity, along with number and type of conditions when managing those with T2D living with multimorbidity. Current guidelines do acknowledge the complex nature of multimorbidity, for which the choice of glycaemic targets and treatment should be based on the patient's individual clinical needs, comorbidities, and the risks from polypharmacy [24]. Therefore, it is important to consider the overall multimorbidity disease burden as a way of recalibrating and personalising our clinical focus in managing people with T2D [25]. However, future studies should aim to explore the mechanisms underpinning the increased mortality and lower HbA1c associated with multimorbidity observed in our findings. This could contribute to better understanding of how to manage specific patterns of multimorbidity and how this should be balanced across the treatment of all multimorbidity conditions. Better understanding of these issues, including effects of condition type, will enable more effective personalisation of care for those with T2D and multimorbidity.

To our knowledge, this is the first study to assess and compare the relationship between total, concordant, and discordant multimorbidity counts; HbA1c; and all-cause mortality in people with T2D. We also show the associations of a wide range of individual conditions included in our multimorbidity counts and mortality. Our study is also novel in that we explored combinations of conditions that were associated with the highest risk of death. Key strengths include use of two national datasets, large sample sizes, recruitment from the UK and Taiwan, and adjustment of our analyses for a wide range of sociodemographic and lifestyle factors. We utilised a robust method to capture multimorbidity conditions using both self-report data and coded hospital data. However, these data were only available at baseline, and we were unable to model for changes in multimorbidity over our study period. Therefore, a limitation of our study is that we were unable to consider the temporality and duration of the conditions in addition to diabetes, which is important for serious conditions such as stroke, TIA, and cancer. UK Biobank is not a random population sample: the participants are more likely to be people of European descent and comparatively less deprived socioeconomically compared with the general UK population [26], suggesting that our findings are likely to be conservative regarding the prevalence of multimorbidity and its associations with mortality. A limitation to note is that despite depression being increasingly common in those with T2D,

there was a comparatively low prevalence of depression in our Taiwan NDCMP cohort. However, this is consistent with evidence suggesting the prevalence of depression is significantly lower in Asia Pacific countries compared with western European countries[27], perhaps because of underdiagnosis of depression due to cultural and social stigma associated with mental health conditions in Asian countries [28]. Although we have classified mental health conditions including depression and anxiety as discordant conditions, there are still debates regarding whether this is appropriate. Studies have shown that depression and anxiety may share biological and behavioural mechanisms [29], which could mean that our classification of these conditions could lead to underestimating the associations between concordant conditions and our outcomes. There was a large overlap between the concordant and discordant condition groups, so a limitation of our study was that for cases in which a person lives with both concordant and discordant conditions, we did not explore this overlap and the individual effects of the two condition groups on HbA1c and mortality. Furthermore, we also did not explore the overlap of the effects of individual conditions within the same condition group; for example, the overlap between chronic liver diseases and alcohol problems. Finally, the fact that the 35 conditions considered in our discordant conditions count included many that are not strong predictors of death may have diluted the overall relationship between discordant condition count and mortality.

In conclusion, increasing multimorbidity is significantly associated with increased mortality in those with T2D and with lower HbA1c. This was observed in two large community cohorts of people from different healthcare systems. The highest risk of mortality is seen in those with concordant conditions, but discordant conditions such as alcohol problems, chronic liver disease, and COPD in UK Biobank and cancer and viral hepatitis in the Taiwan NDCMP cohort were still associated with more than 2-fold the risk of mortality. A key finding is that the combinations of conditions with the greatest association with mortality differed between UK Biobank, a population predominantly comprising people of European descent, and the Taiwan NDCMP, a predominantly ethnic Chinese population. These findings suggest that we need to know more about the influence of different patterns of multimorbidity on outcomes across different ethnic groups in T2D populations. Furthermore, a more cautious approach to tight glycaemic control in some patterns of multimorbidity and T2D may merit consideration. However, further research is needed to explore the mechanisms underpinning these findings. It will be important for clinicians to better understand the biology or healthcare delivery approaches that are contributing to these associations in order to tailor advice to better meet the needs of these diverse and complex populations of people with T2D from different ethnic backgrounds.

## Supporting information

**S1 Text. Study protocol.**
(DOCX)

**S2 Text. STROBE checklist.** STROBE, Strengthening the Reporting of Observational Studies in Epidemiology.
(DOC)

**S1 Table. List of long-term conditions considered for multimorbidity count.**
(DOCX)

**S2 Table. Sensitivity analysis.** Relationship of multimorbidity total count with HbA1c.
(DOCX)

**S3 Table. Sensitivity analysis.** Relationship of multimorbidity total count with all-cause mortality.
(DOCX)

**S4 Table. Sensitivity analysis.** Hospital-verified data.
(DOCX)

**S5 Table. Spline plots of multimorbidity condition count and mortality.**
(DOCX)

## Author Contributions

**Conceptualization:** Jason I. Chiang, Peter Hanlon, Tsai-Chung Li, Bhautesh Dinesh Jani, Jo-Anne Manski-Nankervis, John Furler, Cheng-Chieh Lin, Shing-Yu Yang, Barbara I. Nicholl, Sharmala Thuraisingam, Frances S. Mair.

**Data curation:** Jason I. Chiang, Tsai-Chung Li, Bhautesh Dinesh Jani, Cheng-Chieh Lin, Shing-Yu Yang, Barbara I. Nicholl.

**Formal analysis:** Jason I. Chiang, Peter Hanlon, Bhautesh Dinesh Jani, Shing-Yu Yang, Sharmala Thuraisingam.

**Investigation:** Jason I. Chiang, Jo-Anne Manski-Nankervis, John Furler, Shing-Yu Yang, Barbara I. Nicholl, Sharmala Thuraisingam, Frances S. Mair.

**Methodology:** Jason I. Chiang, Peter Hanlon, Tsai-Chung Li, Bhautesh Dinesh Jani, Jo-Anne Manski-Nankervis, John Furler, Cheng-Chieh Lin, Shing-Yu Yang, Barbara I. Nicholl, Sharmala Thuraisingam, Frances S. Mair.

**Project administration:** Jason I. Chiang.

**Supervision:** Jo-Anne Manski-Nankervis, John Furler, Frances S. Mair.

**Writing – original draft:** Jason I. Chiang.

**Writing – review & editing:** Jason I. Chiang, Peter Hanlon, Tsai-Chung Li, Bhautesh Dinesh Jani, Jo-Anne Manski-Nankervis, John Furler, Cheng-Chieh Lin, Shing-Yu Yang, Barbara I. Nicholl, Sharmala Thuraisingam, Frances S. Mair.

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
