## [Decision Letter · Decision Letter 0]

18 Dec 2019

Dear Dr. Chiang,

Thank you very much for submitting your manuscript "Multimorbidity in Type 2 Diabetes: Associations With Mortality and HbA1c in Two Population Cohorts (UK and Taiwan)" (PMEDICINE-D-19-03306) for consideration at PLOS Medicine. 

[LINK]

In light of these reviews, I am afraid that we will not be able to accept the manuscript for publication in the journal in its current form, but we would like to consider a revised version that addresses the reviewers' and editors' comments. Obviously we cannot make any decision about publication until we have seen the revised manuscript and your response, and we plan to seek re-review by one or more of the reviewers. 

We expect to receive your revised manuscript by Jan 08 2020 11:59PM. Please email us (plosmedicine@plos.org) if you have any questions or concerns.

We look forward to receiving your revised manuscript. 

Sincerely,

Clare Stone, PhD

Managing Editor 

PLOS Medicine

plosmedicine.org

The abstract is written is a slightly ‘listy’ way with headings of outcomes etc. Please remove and write in a more traditional style and avoid short statements such as ‘Baseline HbA1c; all-cause mortality’; as the final sentence of the ‘Methods and findings’ section please add a sentence or 2 on the limitations of the study; please add p values where 95%Cis are given; please add some baseline demographic information on the participants in the cohorts;

At this stage, we ask that you include a short, non-technical Author Summary of your research to make findings accessible to a wide audience that includes both scientists and non-scientists. The Author Summary should immediately follow the Abstract in your revised manuscript. This text is subject to editorial change and should be distinct from the scientific abstract. Please

see our author guidelines for more information: https://journals.plos.org/plosmedicine/s/revising-your-manuscript#loc-author-summary

Square brackets for refs in main text – please place these before the punctuation. And when occurring mid-sentence please insert a space before the open bracket.

The use of MM throughout – as this probably generates lots of returns in PubMed searches please avoid using MM. I also think co-morbidity is a more commonly used phrase? So in any case remove ‘MM’ throughout, please and consider instead co-morbidity (spelt out in full and not as CM). Further to this, in both the abstract and main text on first mention, can you please list what these other conditions are or at least the major ones. 

Please avoid use of quote marks (“”) for emphasis or quoting other papers. 

Line 133 – can you clarify what light ‘do-it-yourself (DIY)’ activity only in the last 4week means? Are you defining exercise for this category as only doing DIY? It reads oddly to say exercise is DIY.

Table 3, please remove ‘present’after diabetes and write as ‘diabetes plus….’ (also for Table 4 and any other similar occurences, please).

Fig 1 – add a time scale (dates) to the X axis in all panels

Main text – please add p values with Cis

As Ref 3 also mentions, please avoid causal language as this is not a trial.

STROBE checklist – thank you for supplying – please also add the paragraph numbers to the sections. 

Comments from the reviewers:

Reviewer #1: I confine my remarks to statistical aspects of this paper. I have a couple general questions:

1. Why treat count independent variables as if they were nominal? Even with the "4 or more" category, it is at least ordinal. Ordinal IVs are a bit tricky, but one method is optimal scaling. In SAS this is avaiable in PROC TRANSREG. In R, there is the optiscale package. But, since you have the actual count, you could treat it as continuous and maybe use a spline to see if there are nonlinearities.

2. Although it isn't necessary, I think it would be very interesting to see if the effects of comorbities are additive or not, at least for the more common examples. That is, is the increased risk of death for people with T2D and CHD equal to the increase from T2D PLUS the increase from CHD? Or is it more? Or less? (This would require a normative sample .... it might even be a different paper).

Specific comments

Line 120 - why group 4 or more? It's OK for tables, but not a great idea for models (see above)

Line 125-126 Don't categorize BMI. Categorizing continuous variables is almost always a mistake. I wrote about this here: https://medium.com/@peterflom/what-happens-when-we-categorize-an-independent-variable-in-regression-77d4c5862b6c?source=friends_link&sk=1428cd15968e218268121dc507ce8025 and here https://medium.com/@peterflom/why-binning-continuous-data-is-almost-always-a-mistake-ad0b3a1d141f?source=friends_link&sk=1d12e016e9dee55f273d38ab6258fb22

 Similar issues for amount of premium in Taiwan

Line 130 - why group current and former smokers?

Line 131-132 - if this is how the data were collected on alcohol, well ... then there isn't much you can do. But surely risk increases a lot at the high end (people who binge driink or drink many drinks per day)

Line 138-139 - same sort of question as above.

Line 143 - maybe I am misunderstanding something, but how is baseline HbA1c an outcome?

Table 1, 2 : Give median and IQR for BMI

 The order of alcohol in table 1 is odd

 For the IQR for duration, give it as 2 numbers (e.g 2 to 10). This is much more informative

Table 3 strikes me as very weird. However, given that the mean HbA1c was 6.8, are any of these differences substantively meaningful? (I know some are significant .... but is 6.75 really different from 6.85 in any important way?

Figures - I would label the curves on the right hand margin with 0, 1, 2, 3, 4+ That seems more readable.

Reviewer #2: This paper examines the mortality and HbA1c in people with type 2 diabetes and multimorbidity. This is a highly relevant research question given the rapidly raising number of people with diabetes and multimorbidity, and the novelty of the topic. 

The strengths of the study include two large cohorts from two countries with two different healthcare systems and different ethnicities, combined with extended follow-up durations. The authors found that increasing total MM and discordant condition counts were associated with slightly lower HbA1c and significant associations between increasing MM and risk of mortality, which was consistent for total count of MM, as well as counts of, concordant and discordant conditions. 

The manuscript is clearly and succinctly written and addresses a novel aspect of a subject of increasing interest and importance, and of notable clinical relevance. Nonetheless, I would say the results are less surprising, I have the following concerns:

1. The conditions for the UK Biobank was ascertained either by self-reported or linked hospital data. This means that other conditions were assessed imprecisely and a large proportion of diabetes cases were missed as the ascertainment method did not include undiagnosed diabetes. It is unclear how this affects on study conclusion. I suggest the author re-run the models by using verified hospital cases to see whether the results were the same as combing both self-reported and hospital verified data.

2. Although the authors adjusted the duration of the diabetes, what about the duration of other conditions? This is very important for some fatal conditions, for example, cancer, stork and TIA, which may take into account the competing risk, that is, the higher mortality risk in patients with diabetes and these conditions was because these conditions were the first occurring ones, and diabetes was a secondary one.

3. Although there was some evidence to support the classification of these conditions into concordant and discordant, it's still controversial for some, for example, depression and anxiety, many studies suggest that these mental disorders and diabetes shared some common pathways. The discussion of this point and the potential resulting limitations should be expanded upon in the "Strengths and limitations" section of the Discussion.

4. The prevalence of concordant and discordant conditions in the UK Biobank were 76.1% and 66.9% separately, there is a huge overlap between the two condition groups. However, it seems the authors did not explain what happened and how to deal with this overlap. If one give person has suffered from both concordant and discordant conditions, can the author figure out the effect on the HbA1c and mortality from which condition group? Furthermore, there are also overlaps among the included discordant conditions, for example, it's obvious that many people with alcohol problems would develop chronic liver diseases, depression and other conditions. It means the effects of alcohol problems may from other conditions.

5. The authors mentioned the importance to look at the effect of multimorbidity patterns in patients with diabetes. Considering the big sample size, can the author explore which pattern is more harmful in patients with diabetes, but not only analysing the counts of conditions, which would be very interesting for clinical practice and disease control and prevention. 

6. Lines 212-214, these results (For a given count of concordant conditions, mortality was higher (or survival lower) than for an equivalent count of discordant conditions or any conditions) were not from the same model, can we compare them in such way?

7. There is a special issue at the journal, it would be valuable to read these papers: https://collections.plos.org/cvd-special-issue

Reviewer #3: Main points / suggestions

The strengths of this paper are the use of two very large datasets and a breakdown of different types of multi morbidity.

Main suggestion - I suggest the authors discuss more the possible reasons behind the apparently paradoxical association with increased MM and lower HbA1c. For example the authors make little mention of survival bias. One possible reason they see consistent associations between more diseases and lower hba1c is that those with more conditions were more likely have died or be too sick to participate in the studies. They describe the mortality effect nicely but don't appear to make the link. This is important to acknowledge because it means some conclusions need to be reworded eg "these findings suggest a more cautious approach to tight glycaemic control in some MM in type 2 diabetes may be warranted." Although they mention another possibilty - better care, but they also do not explicitly mention possible better and earlier diagnosis - for example people with a heart condition will be tested for HbA1c, and so mild HbA1c is more likely to be picked up as well as treated.

other suggestions: 

2. it would be useful to add non diabetic frequencies of conditions and repeat analyses limited to conditions significant (at p<0.05/42 conditions ) more freq in diabetes. This will help the reader assess whether or not some conditions are prsent in people with diabetes by chance, at similar frequuences as the general population. 

3. the tables dont include numbers of deaths. These Ns would be useful to add. 

4. consider using the new HbA1c scale. 

5. Take care with causal language- statements such as "have effects on mortality" should be reworded as "associated with mortality"

6. Take care with multivariable vs multivariate (sensitivity analyses in methods)

[LINK]

---

## [Decision Letter · Decision Letter 1]

18 Feb 2020

Dear Dr. Chiang,

Thank you very much for re-submitting your manuscript "Multimorbidity in Type 2 Diabetes: Associations With Mortality and HbA1c in Two Population Cohorts (UK and Taiwan)" (PMEDICINE-D-19-03306R1) for review by PLOS Medicine.

I have discussed the paper with my colleagues and the academic editor and it was also seen again by xxx reviewers. I am pleased to say that provided the remaining editorial and production issues are dealt with we are planning to accept the paper for publication in the journal.

[LINK]

We look forward to receiving the revised manuscript by Feb 25 2020 11:59PM. 

Sincerely,

Clare Stone, PhD

Managing Editor 

PLOS Medicine

plosmedicine.org

Requests from Editors:

Title- to conform to house style, we suggest a change from Multimorbidity in Type 2 Diabetes: Associations With Mortality and HbA1c in Two Population 2 Cohorts (UK and Taiwan) 

To

Multimorbidity with type 2 diabetes and associations between mortality and HbA1c: a cohort study with UK and Taiwanese cohorts. 

It would be beneficial to emphasize the context a little more, perhaps ie “Evidence suggests that those with more concordant conditions have improved diabetes care due to synergistic care, where diabetes guidelines often make specific recommendations for concordant conditions but do not address discordant conditions [5, 26]”. Please highlight this context a tad more- in the abstract and earlier in the conclusion section.

Page 19- there are a few sentences line 306, 311 and 316 where you say “The

306 HRs for the Taiwan NDCMP were slightly lower yet still statistically significant”. The results are in Table 4 but I wonder why UK biobank results are called out and not the Taiwanese results? Please cite Table 4 in this section a bit more since that's where all the numbers are.

Please add summary demographic information to the abstract. I realise you added age already but can you say how many men, women for example and also things like how many are obese or smokers, etc. 

I would suggest removing DIY. Thank you for the previous clarification around this term. You define it as chores such as mowing the lawn, so I think as DIY is used to define tasks like woodwork and building things, id stick to light or more heavy-duty household or garden tasks. Also at lines 202-205.

In the Author Summary, please remove the text from the author instx, such as “We ask authors to provide 2-3 single sentence bullet points for each of the following questions. 71 Bullet points should be objective, brief, succinct, specific, accurate, and avoid technical language. 72 Why Was This Study Done? Authors should reflect on what was known about the topic before the 73 research was published and why the research was needed.” We only need the actual bullet points, but you have provided too many. We need 2-3 bullet points for each of the 3 sections. 

Line 346 “alcohol problems” please define accurately

There appear to be different fonts in the text – please be consistent (eg line 432)

Comments from Reviewers:

Reviewer #1: The authors have addressed my concerns and I now recommend publication

Peter Flom

Reviewer #3: THe authors have responded well to my suggestions. I would still prefer them to mention in the discussion more explicitly that the apparently paradoxical association between lower HbA1c and higher multimorbidity could be due to surivival or similar collider type biases. The individuals in the UK biobank study had an average age of 58 at the time of HbA1c measurment. At this average age it is likely that people with type 2 diabetes and multiple additional conditions "need" to have less severe diabetes compared to those with no other conditions, to be alive and well enough to participate in the study.

[LINK]

---

## [Editor Report · Decision Letter 2]

10 Apr 2020

Dear Dr. Chiang, 

On behalf of my colleagues and the academic editor, Dr. Ronald C.W. Ma, I am delighted to inform you that your manuscript entitled "Multimorbidity, mortality and HbA1c in type 2 diabetes: a cohort study with UK and Taiwanese cohorts" (PMEDICINE-D-19-03306R2) has been accepted for publication in PLOS Medicine. 

PRODUCTION PROCESS

PRESS

PROFILE INFORMATION

Thank you again for submitting the manuscript to PLOS Medicine. We look forward to publishing it. 

Best wishes, 

Clare Stone, PhD

Managing Editor 

PLOS Medicine

plosmedicine.org